# Instability of the NS1 Glycoprotein from La Reunion 2018 Dengue 2 Virus (Cosmopolitan-1 Genotype) in Huh7 Cells Is Due to Lysine Residues on Positions 272 and 324

**DOI:** 10.3390/ijms22041951

**Published:** 2021-02-16

**Authors:** Eva Ogire, Olivier Diaz, Pierre-Olivier Vidalain, Vincent Lotteau, Philippe Desprès, Marjolaine Roche

**Affiliations:** 1Processus Infectieux en Milieu Insulaire Tropical (PIMIT), Université de La Réunion, Inserm UMR 1187, CNRS 9192, IRD 249, Plateforme CYROI, 97490 Sainte-Clotilde, Ile de La Réunion, France; eva.ogire@univ-reunion.fr; 2Centre International de Recherche en Infectiologie (CIRI), Université de Lyon, Ecole Normale Supérieure de Lyon, Inserm, U1111, CNRS, UMR5308, 69007 Lyon, France; olivier.diaz@inserm.fr (O.D.); pierre-olivier.vidalain@inserm.fr (P.-O.V.); vincent.lotteau@inserm.fr (V.L.)

**Keywords:** arbovirus, flavivirus, dengue virus, nonstructural protein 1, soluble viral protein, multimeric viral protein, recombinant viral protein, hepatoma cells

## Abstract

La Reunion island in the South West Indian Ocean is now endemic for dengue following the introduction of dengue virus serotype 2 (DENV-2) cosmopolitan-I genotype in 2017. DENV-2 infection causes a wide spectrum of clinical manifestations ranging from flu-like disease to severe dengue. The nonstructural glycoprotein 1 (NS1) has been identified as playing a key role in dengue disease severity. The intracellular NS1 exists as a homodimer, whereas a fraction is driven towards the plasma membrane or released as a soluble hexameric protein. Here, we characterized the NS1 glycoproteins from clinical isolates DES-14 and RUN-18 that were collected during the DENV-2 epidemics in Tanzania in 2014 and La Reunion island in 2018, respectively. In relation to hepatotropism of the DENV, expression of recombinant DES-14 NS1 and RUN-18 NS1 glycoproteins was compared in human hepatoma Huh7 cells. We observed that RUN-18 NS1 was poorly stable in Huh7 cells compared to DES-14 NS1. The instability of RUN-18 NS1 leading to a low level of NS1 secretion mostly relates to lysine residues on positions 272 and 324. Our data raise the issue of the consequences of a defect in NS1 stability in human hepatocytes in relation to the major role of NS1 in the pathogenesis of the DENV-2 infection.

## 1. Introduction

Dengue is a mosquito-borne viral disease that has recently evolved into a major public health problem in the South West Indian Ocean (SWIO) region, including the French department La Reunion island, where an epidemic is occurring since 2018 [1,2]. Infection with dengue virus (DENV) serotypes 1 to 4 can cause a wide spectrum of clinical manifestations ranging from dengue fever to severe dengue as dengue shock syndrome or dengue hemorrhagic fever [3,4,5]. Individuals are susceptible to re-infection with different DENV serotypes, and the severity of dengue disease is often associated with secondary infections [4,6,7]. To date, there is an increased risk of severe dengue forms in La Reunion due to a co-circulation of the DENV-1, DENV-2, and DENV-3 [8]. In the absence of antiviral therapies, management of severe dengue is essentially based on supportive therapy and fluid resuscitation. A tetravalent dengue vaccine, *Dengvaxia*, was recently licensed but is only recommended for high-risk epidemic and endemic regions and for individuals having serological evidence of previous dengue virus infection [9,10].

La Reunion island is now endemic for dengue following the introduction of the DENV-2 at the end of 2017. Very little is still known about the biological characteristics of the DENV-2 strains circulating in SWIO. The phylogenetic analysis showed that the DENV-2 isolates from La Reunion patients diagnosed for dengue disease in 2018 (Reunion/2018 strains), including the prototypical RUJul strain used in this study (GenBank accession number MN272404), belong to the cosmopolitan 1 lineage and, more precisely, to the C1-B sub-lineage [1,11]. The Reunion/2018 DENV-2 sequences also clustered with those identified in 2016 in Seychelles. Conversely, the DENV-2 strain D2-K2_RIJ059/Dar es Salaam 2014 (GenBank accession number MG189962) that was isolated from a patient with dengue fever during an outbreak in Dar es Salaam in 2014 (Tanzania; TNZ) belongs to the cosmopolitan C2 sub-lineage [12]. This active co-circulation of the DENV-2 cosmopolitan lineages in SWIO is likely in relation to earlier introductions from India, Pakistan or China [1,11,12,13].

Like other mosquito-transmitted flaviviruses of medical interest such as yellow fever virus, West Nile virus, Japanese encephalitis virus and Zika virus, DENV is a single-stranded positive RNA virus [14,15]. The flavivirus genomic RNA, which is approximatively 11 kb in length, is translated into a single polyprotein that is co- and post-translationally processed by cellular and viral proteases into structural proteins C, prM and E, followed by the nonstructural proteins NS1, NS2AB, NS3, NS4AB, and NS5 [14,15,16]. The viral glycoprotein NS1 (352 amino acids) is the only flavivirus nonstructural protein released in the human bloodstream. The biogenesis of NS1 occurs in the lumen of the endoplasmic reticulum (ER), where NS1 forms a homodimer, which contributes to the formation of viral RNA replication complexes at the early stages of virus replication in the host–cell [17,18]. A fraction of hydrophobic NS1 homo-dimers is driven from the Golgi complex towards the plasma membrane or released as hexameric lipoprotein particles in the extracellular compartment [19,20,21]. Soluble NS1 may play a key role in the pathogenesis of flavivirus-associated diseases [19,22,23,24,25,26]. NS1 antigenemia is currently used as a diagnostic biomarker during the acute phase of symptomatic DENV infection [27,28]. A positive correlation between NS1 antigenemia and severe dengue has been documented [29,30]. However, whether the level of NS1 protein relates to the severity of the DENV infection remains an open issue [31].

Given the central role of NS1 in dengue pathogenesis, we wondered whether the molecular and biological characteristics of the NS1 glycoprotein could differ among the various DENV-2 strains circulating in SWIO. In the present study, DENV-2 isolates RUJul (hereafter named RUN-18), and D2-K2_RIJ059/Dar es Salaam 2014 (hereafter named DES-14) were used as sources of the DENV-2 NS1 glycoproteins. The NS1 glycoprotein is divided into three regions identified as β-roll (NS1-1/30), Wing (NS1-37/162), and β-ladder (NS1-180/352) domains [32]. DENV-2 NS1 protein contains two conserved N-linked glycans at positions NS1-130 (Wing domain) and NS1-207 (β-ladder domain) [33]. Although RUN-18 and DES-14 belong to two different DENV-2 cosmopolitan sub-genotypes, there are only four major amino-acid changes differentiating NS1^RUN-18^ from NS1^DES-14^ (Appendix A). The two amino-acid substitutions at positions NS1-128/131 in the Wing domain might have an impact on the accessibility of N130. The two others at positions NS1-272/324 in the β-ladder domain involve conserved basic residues. To determine whether the four amino-acid changes differentiating RUN-18 NS1 from DES-14 NS1 influence NS1 protein expression, recombinant SWIO DENV-2 NS1 glycoproteins have been expressed in human epithelial HEK-293T and hepatoma Huh7 cells. We observed that RUN-18 NS1 was poorly stable in Huh7 cells compared to DES-14 NS1. By directed mutagenesis, we identify the lysine residues on positions 272 and 324 as playing a key role in RUN-18 NS1 instability.

## 2. Results

### 2.1. Expression of Recombinant SWIO DENV-2 NS1 Proteins in HEK-293T Cells

#### 2.1.1. Antigenic Reactivity of SWIO DENV-2 rNS1 Protein

The SWIO DENV-2 strain DES-14 was used as a reference sequence for NS1 glycoprotein. A synthetic gene with optimized codons for DES-14 NS1 (rNS1^DES-14^) expression in mammalian cells was inserted in the vector plasmid pcDNA3 (Figure 1A). In resulting plasmid pcDNA3/DENV-2.rNS1^DES-14^, the NS1 sequence is preceded by a signal peptide of human secreted protein [34] and ended by spaced FLAG and 6x(His) tags as C-terminal heterologous epitopes for assessing NS1 expression in transfected cells. Then, starting from plasmid pcDNA3/DENV-2.rNS1^DES-14^, directed mutagenesis was used to generate the pcDNA3/DENV-2.rNS1^RUN-18^ coding for RUN-18 NS1 (rNS1^RUN-18^) (Figure 1B).

Antigenic reactivity of rNS1^DES-14^ (380 amino acids, including the C-terminal spacers and tags) was assessed on HEK-293T cells transfected 24 h with pcDNA3/DENV-2.rNS1^DES-14^. Cell lysates in RIPA buffer were analyzed by immunoblotting assay (Figure 2). Given that newly synthesized NS1 protein is rapidly converted to a heat-labile dimeric form in the ER compartment [35], cell lysates were analyzed before and after heat-denaturation in the presence of a reducing agent. Antibodies raised against FLAG and 6x(His) epitopes were able to detect both monomeric (apparent MW estimated to 45 kDa), and dimeric (apparent MW estimated to 100 kDa) forms of rNS1^DES-14^ expressed in HEK-293T cells. The antigenic reactivity of rNS1^DES-14^ was verified with a serum from a symptomatic individual diagnosed for the DENV-2 infection in La Reunion in 2018 (Figure 2). We noted that patient serum displayed a greater reactivity with rNS1^DES-14^ dimeric form as compared to the rNS1^DES-14^ monomeric form. These results demonstrated the antigenic reactivity of rNS1^DES-14^ expressed in HEK-293T cells.

#### 2.1.2. Expression of SWIO DENV-2 NS1 Monomeric and Dimeric Forms

We then investigated the biological impact of the four amino-acid changes discriminating rNS1^RUN-18^ from rNS1^DES-14^. Site-directed mutagenesis was performed on pcDNA3/DENV-2.rNS1^DES-14^ to generate a recombinant plasmid encoding rNS1^RUN-18^ (pcDNA3/DENV-2.rNS1^RUN-18^). The resulting rNS1^RUN-18^ sequence differs from rNS1^DES-14^ by the amino-acid substitutions L128P, H131Q, R272K, and R324K (Figure 1B). Expression of rNS1^RUN-18^ was assessed in HEK-293T cells transfected 24 h with pcDNA3/DENV-2.rNS1^RUN-18^. As observed for rNS1^DES-14^ (Figure 2), immunoblot assay on RIPA cell lysates using anti-FLAG antibody allowed the detection of both rNS1^RUN-18^ monomeric and dimeric forms in HEK-293T cells (Figure 3).

We then evaluated the impact of the four amino-acid substitutions that differentiate rNS1^RUN-18^ from rNS1^DES-14^ on the conformation of recombinant DENV-2 NS1 glycoprotein. Immunoblot assays were performed with a mouse monoclonal anti-flavivirus NS1 antibody (D/2/D6/B7) (NS1 mAb^D/2/D6/B7^), which recognizes a conformational epitope, and a rabbit DENV-2 NS1 polyclonal antibody PA5-32207 (NS1 Ab^PA5-32207^), which recognizes linear epitopes in the NS1 region 117–301. As expected, NS1 mAb^D/2/D6/B7^ detected rNS1 dimeric form into native samples, whereas NS1 Ab^PA5-32207^ recognized rNS1 monomeric form in heat-denatured samples. Surprisingly, we observed a weak antigenic reactivity of rNS1^RUN-18^ in relation to NS1 mAb^D/2/D6/B7^ and NS1 Ab^PA5-32207^ as compared to rNS1^DES-14^ (Figure 3). Such a result suggests that the rNS1^RUN-18^ glycoprotein structure was impacted by the four amino-acid changes at positions NS1-128/131/272/324.

### 2.2. Expression of Recombinant SWIO DENV-2 NS1 Proteins in Huh7 Cells

#### 2.2.1. Expression of DES-14 rNS1 and RUN-18 rNS1 Proteins

To investigate whether the four amino-acid substitutions differentiating rNS1^RUN-18^ from rNS1^DES-14^ influence DENV-2 NS1 glycoprotein expression in human host cells for the DENV such as hepatocytes, Huh7 cells were transfected 24 h with pcDNA3/DENV-2.rNS1^RUN-18^ or pcDNA3/DENV-2.rNS1^DES-14^. Expression of rNS1 in transfected Huh7 cells was first assessed by immunofluorescence analysis using anti-6x(His) antibody (Appendix A). Comparable fractions of rNS1 positive cells were observed after transfection with pcDNA3/DENV-2.rNS1^RUN-18^ or pcDNA3/DENV-2.rNS1^DES-14^, thus allowing comparative analysis in Huh7 cells (Appendix A).

Immunoblot assays were then performed on RIPA cell lysates using the two anti-NS1 antibodies (Figure 4A). Anti-FLAG antibody was used to detect both rNS1 monomeric and dimeric forms. The use of anti-NS1 pAb PA5-32207 allowed the strong detection of rNS1^DES-14^ monomeric form in Huh7 cells, but a much lower signal was observed for rNS1^RUN-18^ monomeric form. Both anti-NS1 mAb ^D/2/D6/B7^ antibody and anti-FLAG antibody detected a weak amount of rNS1^RUN-18^ dimeric form in Huh7 cells as compared to rNS1^DES-14^ (Figure 4A). We determined relative abundances of rNS1 proteins in immunoblot assays (Figure 4B). The quantification of intensity signals revealed that amounts of rNS1^RUN-18^ monomeric and dimeric forms represent only 20% of their rNS1^DES-14^ counterparts. This suggests that the four NS1-128/131/272/324 residues may have an impact on the stability of rNS1^RUN-18^ expressed in Huh7 cells.

#### 2.2.2. Impact of Mutations on rNS1^RUN-18^ Expression in Huh7 Cells

To understand the role of the four amino acid substitutions NS1-128/131/272/324 on the structural change of rNS1^RUN-18^ in Huh7 cells, site-directed mutagenesis was performed on pcDNA3/DENV-2.rNS1^RUN-18^ to introduce amino-acid substitutions P128L/Q131H and K272R/K324R into the rNS1^RUN-18^ sequence (Figure 5A). The resulting plasmids pcDNA3/DENV-2.rNS1^RUN-18^-(L128, H131) and pcDNA3/DENV-2.rNS1^RUN-18^-(R272, R324) were used to express the rNS1^RUN-18^ mutants in Huh7 cells. Expression of rNS1^RUN-18^-(L128, H131) and rNS1^RUN-18^-(R272, R324) mutants was verified by IF assay using anti-6x(His) antibody (Figure 5B).

Expression of rNS1^RUN-18^ mutants was compared to rNS1^RUN-18^ and rNS1^DES-14^ in Huh7 cells by immunoblot assay (Figure 4). We showed that the two anti-NS1 antibodies, as well as the anti-FLAG antibody, detected both rNS1^RUN-18^-(L128, H131) and rNS1^RUN-18^-(R272, R324) in Huh7 cells (Figure 4A). The amino-acid substitutions P128L and Q131H resulted in a moderate increase in the amount of the rNS1^RUN-18^ monomeric form detected in Huh7 cells (Figure 4B). Anti-FLAG antibody labeling showed comparable amounts of rNS1^RUN-18^-(R272, R324) mutant and rNS1^DES-14^ in Huh7 cells. A greater antigenic reactivity was observed with the monomeric form of rNS1^RUN-18^-(R272, R324) mutant as compared to rNS1^RUN-18^ monomeric form in relation to anti-NS1 pAb PA5-32207 (Figure 4B). These results suggested that rNS1^RUN-18^ is impacted by the R272 and R324 residues and, to a lesser extent, the L128 and H131 residues. Analysis of rNS1 dimeric forms revealed that rNS1^RUN-18^-(R272, R324) mutant and rNS1^DES-14^ had a comparable pattern of expression, whereas rNS1^RUN-18^-(L128, H131) mutant was intermediate between rNS1^RUN-18^ and rNS1^DES-14^ (Figure 4B). Taken together, the results suggest that instability of rNS1^RUN-18^ expressed in Huh7 cells mostly relate to lysine residues on positions 272 and 324.

#### 2.2.3. The P128 and Q131 Residues Have No Influence on rNS1^RUN-18^ Glycosylation

Our previous data suggested that amino-acid substitutions P128L and Q131H could have an impact on rNS1^RUN-18^ expression in Huh7 cells (Figure 4). Given that the P128 and Q131 residues in the Wing domain are located upstream and inside the first N-glycosylation site of rNS1^RUN-18^, we wondered whether these two residues influence glycan–protein linkage of rNS1. To evaluate the migration profile of rNS1^RUN-18^ in relation to N-linked protein glycosylation, a non-glycosylated rNS1^RUN-18^ mutant was generated by site-directed mutagenesis. It is expected that mutations N130Q and N207Q result in a complete lack of glycan linked to the DENV-2 rNS1 protein. Consequently, amino-acid change Asn-to-Glu was introduced at positions NS1-130 and NS1-207 of rNS1^RUN-18^ in order to generate the rNS1^RUN-18^-(Q130, Q207) mutant (Figure 6A). As a control, we engineered a rNS1^DES-14^ -(Q130, Q207) mutant. The expression of rNS1^DES-14^-(Q130, Q207) and rNS1^RUN-18^-(Q130, Q207) mutants in Huh7 cells was verified by immunofluorescence at 24 h post-transfection using anti-6x(His) antibody (Figure 6B).

Immunoblot assay was performed on RIPA cell lysates using anti-NS1 mAb ^D/2/D6/B7^ (Figure 6C). The detection of the non-glycosylated dimeric form of rNS1 in Huh7 cells indicates that the DENV NS1 dimerization can occur in the absence of N-linked glycosylation [36]. Both rNS1^DES-14^-(Q130, Q207) and rNS1^RUN-18^-(Q130, Q207) dimeric forms migrated faster than rNS1^DES-14^ or rNS1^RUN-18^ consistent with a lack of glycan–protein linkage on rNS1. Analysis of the DENV-2 rNS1 glycoproteins and their non-glycosylated mutants revealed no change in the migration profile between rNS1^DES-14^ and rNS1^RUN-18^ dimers (Figure 6C). Thus, it seems unlikely that the P128 and Q131 residues influence occupancy of the first N-glycosylation site in the Wing domain of rNS1^RUN-18^.

#### 2.2.4. Secretion of rNS1^RUN-18^ Is Impacted by the NS1-128/131/272/324 Residues

To determine whether the NS1-128/131/272/324 residues may have an effect on the secretion of soluble rNS1^RUN-18^ in human hepatoma cells, Huh7 cells were transfected with plasmids expressing rNS1^RUN-18^ or the related mutants rNS1^RUN-18^-(L128, H131) and rNS1^RUN-18^-(R272, R324). At 48 h post-transfection, culture supernatants from Huh7 cells were collected and analyzed by a dot-blot assay using anti-FLAG antibody or anti-NS1 mAb D/2/D6/B7 to quantify the presence of rNS1 (Figure 7A). The signal intensity of immunolabeled rNS1 was determined (Figure 7B). The secretion level of rNS1^RUN-18^ represented only 20% of rNS1^DES-14^ in line with the instability of rNS1^RUN-18^ in Huh7 cells (Figure 4). There was a two-fold higher level of secreted rNS1^RUN-18^-(L128, H131) mutant compared to rNS1^RUN-18^, whereas comparable amounts of rNS1^RUN-18^-(R272, R324) and rNS1^DES-14^ were released by the Huh7 cells. Thus, the efficacy of rNS1^RUN-18^ release in Huh7 cells is strongly affected by the NS1-128/131/272/324 residues. Only amino-acid changes K272R and K324R have the capability to increase rNS1^RUN-18^ secretion at a level similar to rNS1^DES-14^.

We wondered whether the efficient release of soluble rNS1^DES-14^ in Huh7 cell culture supernatants was the consequence of a greater loss of cell viability than rNS1^RUN-18^. Consequently, Huh7 cells were transfected with plasmids expressing either rNS1^DES-14^ or rNS1^RUN-18^ and its mutants. The effect of rNS1 on cell metabolism at 48 h post-transfection was evaluated first by measuring MTT activity, which reflects mitochondrial respiration. Expression of rNS1^DES-14^ resulted in a significant decrease in cell metabolism activity, whereas rNS1^RUN-18^ had no significant effect (Figure 8A). MTT activity was significantly reduced in Huh7 cells expressing rNS1^RUN-18^-(L128, H131) and rNS1^RUN-18^-(R272, R324) mutants (Figure 8A). Given that both rNS1^RUN-18^-(L128, H131) and rNS1^RUN-18^-(R272, R324) mutants have a moderate impact on cell metabolism, a severe loss of mitochondrial respiratory control necessitates the L128/H131/R272/R324 residues present in rNS1^DES-14^. However, this was not associated with higher levels of cellular lysis in response to rNS1^DES-14^ expression. Indeed, when quantifying lactate dehydrogenase (LDH) release in culture supernatants of Huh7 cells expressing either rNS1^DES-14^ or rNS1^RUN-18^, a comparable level of LDH activity was observed (Figure 8B). We can conclude that the efficient release of soluble rNS1^DES-14^ in Huh7 cells did not relate to a greater loss of cell viability. We noted that cellular metabolism is impacted by rNS1^DES-14,^ but not rNS1^RUN-18,^ in line with a high-level of rNS1^DES-14^ expression in Huh7 cells.

## 3. Discussion

The SWIO countries are characterized by the presence of several islands. The French island La Reunion was a non-endemic region for dengue until the first epidemic of dengue serotype 2 in 2018 due to the introduction of the DENV-2 cosmopolitan genotype [1]. Today, La Reunion island experiences a major dengue epidemic where three serotypes are co-circulating [8]. Epidemiological data from recent years revealed a predominant spread of the DENV-2 cosmopolitan lineage in the Asian continent [37,38,39,40,41]. Based on epidemiology investigation and genetic analysis of circulating viral strains in SWIO, it is most likely that the recent DENV-2 outbreak in La Reunion island originates in viral strains imported from the Republic of Seychelles. DENV-2 strain RUNJul (also named RUN-18) was isolated from an autochthonous patient diagnosed with mild dengue disease in La Reunion in 2018 [1]. The sequencing of RUN-18 revealed a close phylogenetic relationship with isolates of the DENV-2 cosmopolitan genotype, which were identified in Seychelles in 2016 [1,11].

Very little information is available on the biological characteristics of circulating DENV-2 cosmopolitan genotype in SWIO. Given that NS1 glycoprotein may play a central role in the pathogenesis of severe dengue disease [19,22,23,24,25,26], we decided to characterize the biological properties of the RUN-18 NS1 protein. A close genetic relationship was identified between the NS1 genes from RUN-18 and DENV-2 strain D2-K2_RIJ059/Dar es Salaam 2014 (also named DES-14) isolated from a dengue patient in Tanzania during a dengue outbreak in 2014 (Appendix A) [12]. There are only four amino-acid residues that differentiate the NS1 sequences of RUN-18 and DES-14. The amino-acid changes involve the residues 128/131/272/324 with the first group of mutations at positions 128/131 in the Wing domain and a second one at positions 272/324 in the β-ladder domain. In relation to hepatotropism of the DENV-2, the RUN-18 NS1 and DES-14 NS1 genes were inserted into a vector plasmid, and protein expression was assessed in transfected human hepatoma Huh7 cells.

The major finding of our study was the weak stability of RUN-18 NS1 in Huh7 cells as compared to DES-14 NS1. An effort was made to evaluate the impact of four amino-acid changes differentiating RUN-18 NS1 and DES-14 NS1 on the stability of the DENV-2 NS1 glycoprotein. Both RUN-18 and DES-14 NS1 glycoproteins comprise two Asn-x-Thr sequons with glycans linked to N130 (Wing domain) and N207 (β-ladder domain) (Appendix A). The Wing domain encompasses the NS1-128/130 residues [33]. In the first Asn-x-Thr sequon of NS1, x is a Gln in RUN-18 NS1 but is a His in NS1 of DES-14 and GZ40, which corresponds to a shift from a neutral to a basic polar residue (Appendix A). The amino-acid substitution P128L maps in the close vicinity of N130 with a Pro residue in RUN-18 NS1 glycoprotein (also detected in the DENV-2/PK/2010), but a Leu residue in DES-14 NS1 (also detected in GZ40) (Appendix A). This suggested that a Pro residue could have an impact on the secondary structure of the N-x-T sequon with consequences on the linkage of glycans to N130. Analysis of RUN-18 NS1 and DES-14 NS1 glycoproteins in Huh7 cells has not allowed us to observe any change in their glycan–protein linkage. However, the amino-acid substitutions P128L and Q131H have a minor effect on RUN-18 NS1 expressed in Huh7 cells. The impact of the P128 and Q131 residues on the stability of RUN-18 NS1 glycoprotein remains to be understood.

Our data demonstrated that only the amino-acid substitutions K272R and K324R have the ability to increase RUN-18 NS1 stability and secretion, resulting in a loss of metabolic activity in Huh7 cells comparable to what is observed with DES-14 NS1. We consider that the Lys residues at positions 272 and 324 contribute to the instability of RUN-18 NS1 glycoprotein expressed in Huh7 cells. The K272 and K324 residues were detected in NS1 proteins from Seychelles 2016 and La Reunion 2018 clinical isolates of the DENV-2 cosmopolitan-I genotype. We noted that NS1 proteins from the DENV-2 strains D2-K2_RIJ059/Dar es Salaam 2014 (DES-14) isolated in Tanzania in 2014 and GZ40 in China in 2010 have Arg at residues 272 and 374 (Appendix A). Analysis of different DENV-2 NS1 glycoproteins usually identified a Lys residue at position 272 and Arg at position 324. This was verified with various isolates of the DENV-2 cosmopolitan genotype such as viral strains collected in Pakistan in 2010 (Appendix A), in East Africa in 2016 (GenBank reference number BAU45374), in India in 2013 (KJ018750), in North America in 2016 (KX702404), and in South America in 2016 (MH215277). It seems, therefore, uncommon to identify a Lys residue on both NS1 residues 272 and 324.

Since the instability of RUN-18 NS1 glycoprotein mostly relates to the lysine residues on positions 272 and 324, it is now of priority to understand the mechanisms by which the K272 and K324 residues influence DENV-2 NS1 glycoprotein in human hepatocytes. The instability of RUN-18 NS1 could be due to a post-translational modification through a protein–protein interaction-dependent mechanism. The ubiquitin–proteasome system involving covalent modification of a lysine residue has been proposed for the DENV NS1 protein [42,43]. A hypothesis is that the lysine residues on positions 272 and 324 act as two potential sites of ubiquitin ligation, which target RUN-18 NS1 for proteasome-mediated degradation leading to a defect in NS1 accumulation in Huh7 cells [44,45]. Interestingly, sequence analysis around the RUN-18 NS1 residue K272 identified the tetrapeptide GKLE as a possible SUMO-interaction motif ψKxE where ψ represents a hydrophobic residue and x any residue (Appendix A) [46,47]. Given that SUMOylation has been proposed to play a role in the DENV replication [48], the possibility that RUN-18 NS1, but not DES-14 NS1, contains a SUMO-interaction motif in the β-ladder domain represents an interesting hypothesis. Further studies are required to determine whether the lysine residues on positions 272 and 324 act as substrates for NS1 ubiquitination as well as SUMOylation.

It is now well established that soluble NS1 may play a critical role in the pathogenesis of the DENV infection [19,22,23,24,25,26]. DENV-2 cosmopolitan-1 genotype responsible for the La Reunion island epidemic in 2018 was presumably introduced from Seychelles islands in SWIO. This is the first study to our knowledge on the biological characteristics of NS1 glycoprotein from the DENV-2 cosmopolitan genotype in SWIO. Expression of recombinant DENV-2 NS1 glycoproteins in human hepatoma cells allowed us to identify a key role for the lysine residues on positions 272 and 324 in the instability of RUN-18 NS1. Future studies will investigate whether a single Lys-to-Arg change at position 272 or 324 may have an impact on DENV-2 NS1 stability. In view of the essential roles of NS1 extracellular forms in the pathogenesis of severe dengue, understanding the consequences of the instability of the DENV-2 NS1 glycoprotein in human hepatocytes is an important issue that will be the subject of further investigation.

## 4. Materials and Methods

### 4.1. Cell Lines and Antibodies

Human embryonic kidney HEK-293T cells (ATCC, CRL-1573) were cultured in minimum essential media (MEM) supplemented with 5% heat-inactivated fetal bovine serum (FBS) and human hepatoma Huh7 cells were cultured in DMEM medium supplemented with 10% heat-inactivated fetal bovine serum (FBS, Dutscher, Brumath, France). Both cell lines were cultured with antibiotics (PAN Biotech Dutscher, Brumath, France) at 37 °C under a 5% CO_2_ atmosphere. Mouse anti-DDDDK tag monoclonal antibody (FLAG Ab) and mouse anti-6x(His) monoclonal antibody was purchased from Abcam (Cambridge, UK). The mouse anti-flavivirus NS1 antibody [D/2/D6/B7] (NS1 mAb^D/2/D6/B7^), which reacts with a conformational epitope present on DENV-2, was purchased from Abcam (Cambridge, UK). The rabbit dengue virus type 2 NS1 polyclonal antibody PA5-33207 (NS1 Ab^PA5-32207^) raised against the residues NS1-117/301 was purchased from (Thermo Fisher Scientific, Les Ulis, France). The rabbit anti-β actin polyclonal antibody was purchased from ABclonal (Massachusetts, USA). Goat anti-mouse IgG secondary antibody and Alexa Fluor Plus 488, was purchased from Invitrogen (Thermo Fisher, Les Ulis, France). Anti-mouse and anti-rabbit IgG HRP-conjugated secondary antibodies were purchased from Abcam (Cambridge, UK). DAPI was purchased from Euromedex (Souffelweyersheim, France).

### 4.2. Vector Plasmids Expressing Recombinant DENV-2 NS1 Proteins and Their Mutants

A mammalian codon-optimized gene coding for the mGluR5 signal peptide, followed by the residues NS1-1/352 of the DENV-2 strain D2-K2_RIJ059/Dar es Salaam 2014 (NS1^DES-14^, GenBank access number MG189962) (Appendix A) and ended by the two C-terminal tags FLAG and 6×(His) epitopes spaced by Gly-Ser spacers, were synthesized by GeneCust (Boynes, France) (Figure 1A). A Kozak consensus sequence for initiation of translation was inserted at the 5′end of the synthetic gene. The synthetic gene was cloned into the Nhe I and Not I restriction sites of the pcDNA3.1-Hygro to generate recombinant plasmid pcDNA3/DENV-2.NS1^DES-14^ by GeneCust (Boynes, France). Direct mutagenesis on recombinant plasmid pcDNA3/DENV-2.NS1^DES-14^ to generate pcDNA3/DENV-2.NS1^RUN-18^ or its mutants pcDNA3/DENV-2.NS1^RUN-18^-(L128, H131) and pcDNA3/DENV-2.NS1^RUN-18^-(R272, R324) was performed by GeneCust (Boynes, France). The production of pcDNA3/DENV-2.NS1^DES-14^-(Q130, Q207) and pcDNA3/DENV-2.NS1^RUN-18^-(Q130, Q207) mutants were performed by GeneCust (Boynes, France). The sequencing of plasmid DNA was performed by GeneCust (Boynes, France), and the complete sequence was verified for each recombinant plasmid by the Sanger method. The production of endotoxin-free plasmids pcDNA3/DENV-2.NS1 and the quantification of plasmid DNA were performed by GeneCust (Boynes, France). Cells were transfected with plasmids using lipofectamine 3000 (Thermo Fisher Scientific, Les Ulis, France) according to the manufacturer’s instructions.

### 4.3. Immunofluorescence Assay

Cells monolayers grown on glass coverslips were incubated for 20 min with methanol at −20 °C. Cells were incubated with primary 6x(His) antibody at dilution 1:200 in PBS containing 1% bovine serum albumin (BSA). Goat anti-mouse Alexa Fluor 488 IgG was used as a secondary antibody at the dilution 1:2000, and the nucleus was stained with DAPI. Vectashield reagent (Vector Labs, Premanon, France) was used for the mounting of the glass coverslips. A Nikon Eclipse E2000-U microscope was used for the visualization of the fluorescence. The capture of the fluorescent signal was allowed with a Hamamatsu ORCA-ER camera coupled to the imaging software NIS-Element AR (Nikon, Champigny-sur-Marne, France).

### 4.4. Immunoblot Assay

Cell lysates were performed in RIPA lysis buffer (Sigma, Lyon, France). Proteins were separated by 4–12% SDS–PAGE and transferred into a nitrocellulose membrane. After blocking of the membrane for 1 h with 90% FBS or 5% milk in TBS-Tween, blots were incubated with primary antibody at dilution 1:200. Anti-mouse or anti-rabbit IgG HRP-conjugated secondary antibodies were used at 1:5000 dilution. For dot blot assays, samples were directly loaded on a nitrocellulose membrane and then probed with the primary antibody and then anti-mouse or anti-rabbit IgG HRP-conjugated secondary antibody. The membranes were developed with Pierce ECL Western blotting substrate (Thermo Fisher Scientific, Les Ulis, France) and exposed on an Amersham imager 680 (GE Healthcare). The signal intensity of rNS1 (intracellular monomeric and dimeric forms and soluble forms) was measured by Image J software. The results are the mean of two or three independent assays. The protein abundance ratio between rNS1 and β-actin was determined for each monomeric and dimeric forms.

### 4.5. Cytotoxicity Assays

For lactate dehydrogenase (LDH) assay, cells were seeded in 12-well culture plates. Cytotoxicity was evaluated by quantification of lactate dehydrogenase (LDH) release in cell cultures using CytoTox 96 nonradioactive cytotoxicity assay (Promega, Charbonnières-les-Bains, France) according to the manufacturer’s instructions. The absorbance of converted dye was measured at 490 nm with background subtraction at 690 nm. For the MTT assay, cells were seeded in a 96-well culture plate. Cell monolayers were rinsed with PBS and incubated with a culture growth medium mixed with 5 mg. mL^−1^ MTT (3-[4,5-dimethylthiazol-2-yl]-2,5-diphenyltetrazolium bromide) solution for 1 h at 37 °C. MTT medium was removed, and the formazan crystals were solubilized with dimethyl sulfoxide (DMSO). Absorbance was measured at 570 nm with background subtraction at 690 nm.

### 4.6. Statistical Analysis

Statistical analysis for comparison studies were performed using GraphPad Prism software (version 9, GraphPad software, San Diego, CA, USA). Values of independent experiments were analyzed by one-way ANOVA using Dunnett’s multiple comparisons test or unpaired *t*-test. Values of *p* < 0.05 were considered statistically significant.

## Figures and Tables

**Figure 1 ijms-22-01951-f001:**
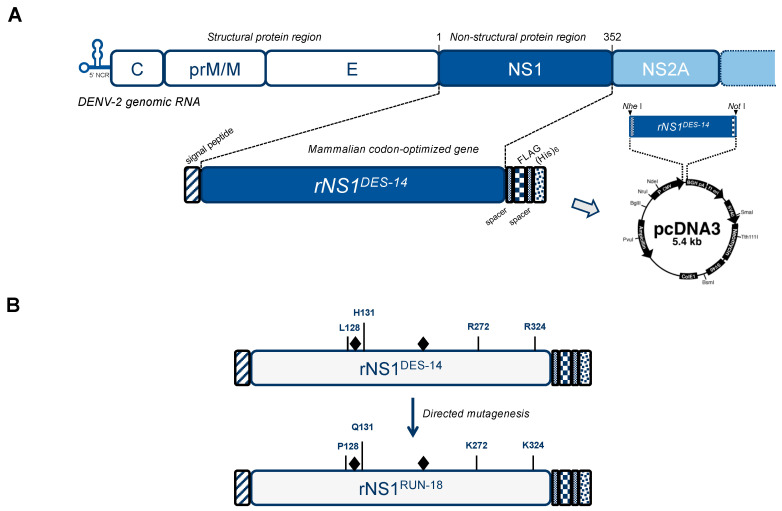
Schematic representation of the dengue virus 2 nonstructural glycoprotein 1 (NS1) constructs. In (**A**), the top panel represents dengue virus serotype 2 (DENV-2) genomic RNA with the structural protein region followed by the nonstructural protein region starting with the NS1 gene. The bottom panel is a schematic representation of the mammalian-codon optimized gene coding for the DENV-2 NS1 protein. The recombinant NS1 construct is preceded by the mGluR5 protein signal peptide (hatched blue box) and ended by spaced FLAG (blue mosaic box) and 6× (His) epitope tags (dotted blue box). The recombinant DENV-2 NS1 gene is inserted into the expression vector pcDNA3. In (**B**), schematic representation of the recombinant NS1 constructs from cosmopolitan Indian Ocean DENV-2 strains D2_K2_RIJ_059 (rNS1^DES-14^) and RUJul (rNS1^RUN-18^). The gray diamonds mention the N-linked glycans linked to N130 and N207. The four amino-acid substitutions differentiating rNS1^RUN-18^ from rNS1^DES-14^ are indicated. The rNS1^RUN-18^ protein has a Val residue at position 84.

**Figure 2 ijms-22-01951-f002:**
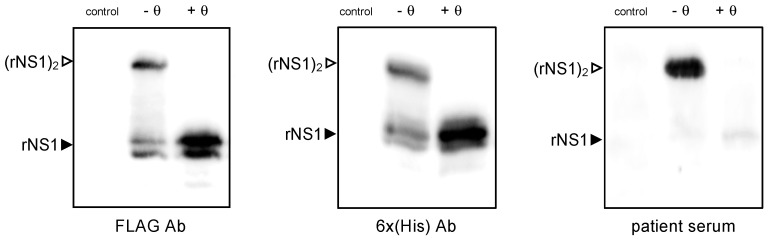
Antigenic reactivity of rNS1^DES-14^. HEK-293T cells were transfected 24 h with recombinant plasmid pcDNA3/DENV-2.rNS1^DES-14^ or mock-transfected (control). Immunoblot assays were performed on RIPA cell lysates. Samples were heat-denatured (+ θ) or not (− θ) and then analyzed by immunoblotting using anti-FLAG antibody (FLAG Ab), anti-6×(His) antibody (6×(His)Ab), or a DENV-2 patient serum (patient serum). The closed arrowheads indicate NS1 monomers (rNS1). The open arrowheads indicate NS1 dimers ((rNS1)_2_).

**Figure 3 ijms-22-01951-f003:**
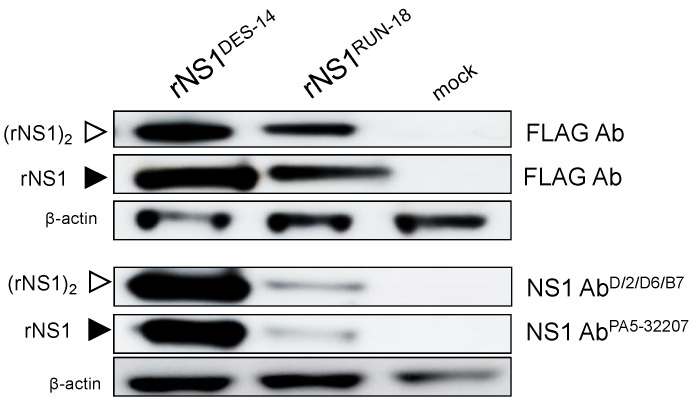
Expression of recombinant DENV-2 NS1 proteins in HEK-293T cells. HEK-293T cells were transfected 24 h with pcDNA3 plasmids expressing rNS1^DES-14^ or rNS1^RUN-18^ (2.5 µg DNA per 10^6^ cells) or mock-transfected (mock). Immunoblot assays on RIPA cell lysates were performed on non-heated samples for detection of NS1 dimeric forms or heated samples for detection of NS1 monomeric forms using anti-FLAG antibody (FLAG Ab), mouse anti-flavivirus NS1 monoclonal antibody D/2/D6/B7 (NS1 Ab^D/2/D6/B7^) or rabbit anti-DENV-2 NS1 polyclonal antibody PA5-32207 (NS1 Ab^PA5-32207^). The closed arrowheads indicate NS1 monomeric form (rNS1). The open arrowheads indicate NS1 dimeric form ((rNS1)_2_). The β-actin protein served as protein-loading control.

**Figure 4 ijms-22-01951-f004:**
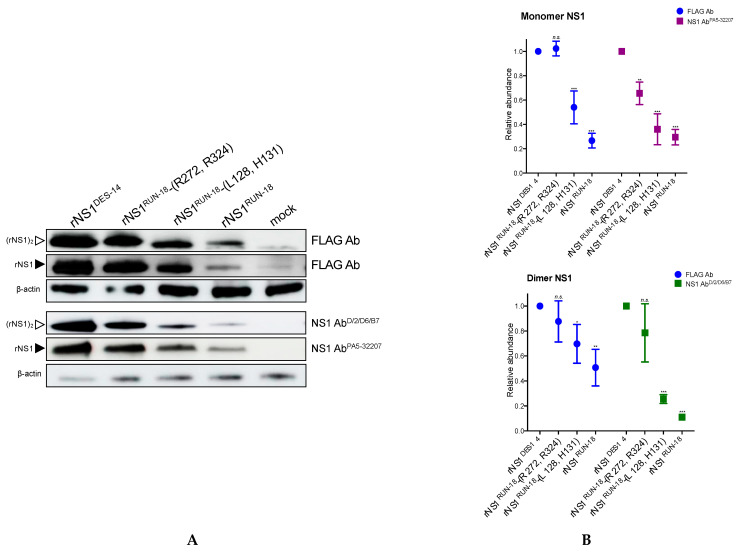
Expression of recombinant SWIO DENV-2 NS1 proteins in Huh7 cells. Huh7 cells were transfected 24 h with recombinant plasmids expressing rNS1^DES-14^, rNS1^RUN-18^, rNS1^RUN-18^ mutants or mock-transfected (mock). In (**A**), a representative immunoblot assay was performed on RIPA cell lysates. Assays were performed on non-heated samples using anti-FLAG antibody (FLAG Ab), or anti-NS1 mAb D/2/D6/B7 (NS1 Ab^D/2/D6/B7^) and heated samples using anti-FLAG antibody (FLAG Ab) or anti-NS1 pAb PA5-32207 (NS1 Ab^PA5-32207^). The closed arrowheads indicate NS1 monomeric forms (rNS1). The open arrowheads indicate NS1 dimeric forms ((rNS1)_2_). The β-actin protein served as a protein-loading control. In (**B**), signal intensities for rNS1 monomeric and dimeric forms were measured using the Image J software. Protein abundance ratios between rNS1 and β-actin were determined for each monomeric and dimeric forms. The abundance of rNS1^RUN-18^ wild-type and mutants is relative to rNS1^DES-14^. The results are the mean of two or three independent assays. *p*-values were determined by comparing rNS1^RUN-18^ wild-type or mutant to rNS1^DES-14^ (*** *p* < 0.001, ** *p* < 0.01, * *p* < 0.1; *n.s.*: not significant).

**Figure 5 ijms-22-01951-f005:**
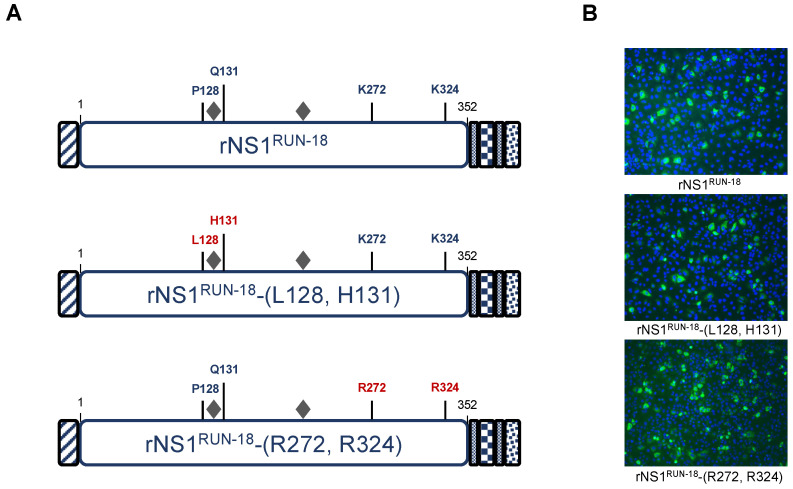
Expression of rNS1^RUN-18^ mutants in Huh7 cells. In (**A**), schematic representation of rNS1^RUN-18^ and related mutants. The amino-acid substitutions in rNS1^RUN-18^ mutants are indicated in red. The gray diamonds mention the N-linked glycans at positions N130 and N207. In (**B**), Huh7 cells were transfected with plasmids expressing rNS1^RUN-18^ or mutants bearing amino-acid substitutions P128L/Q131H (rNS1^RUN-18^-(L128, H131)) or K272R/K324R (rNS1^RUN-18^-(R272, R324)) (2.5 µg DNA per 10^6^ cells). After 24 h of culture, immunofluorescence assays were performed using anti-6x(His) antibody (green) as the primary antibody. Nuclei were stained with DAPI (blue). Immunolabelled cells were visualized by fluorescence microscopy. The same magnification was used throughout (×100).

**Figure 6 ijms-22-01951-f006:**
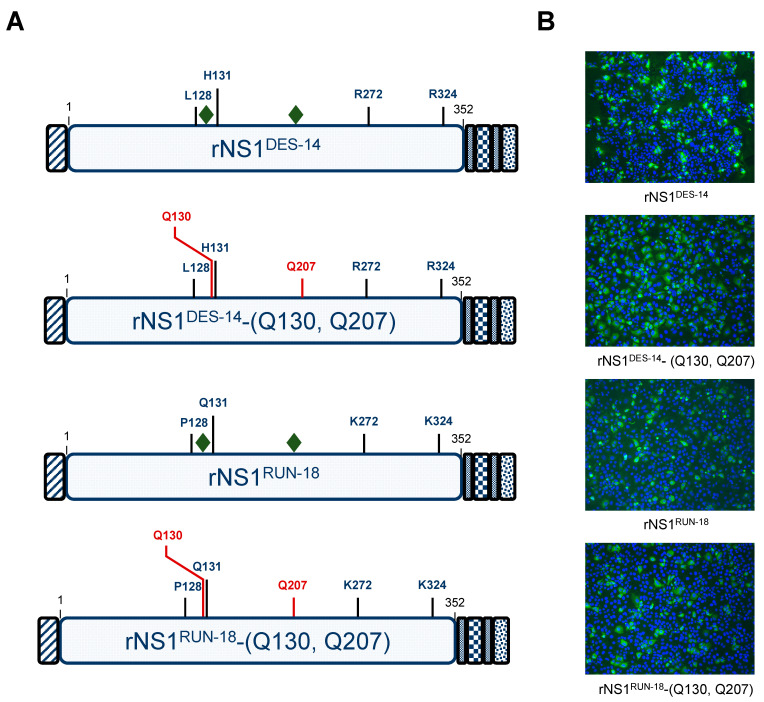
Expression of non-glycosylated recombinant DENV-2 proteins in Huh7 cells. Huh7 cells were transfected 24 h with plasmids expressing rNS1^DES-14^, rNS1^RUN-18^ and their mutants bearing the amino-acid substitutions N130Q and N207Q (rNS1^DES-14^-(Q130, Q207) and rNS1^RUN-18^-(Q130, Q207)) or mock-transfected (mock). In (**A**), schematic representation of rNS1^DES-14^, rNS1^RUN-18^ and their non-glycosylated mutants. Amino-acid substitutions in rNS1^DES-14^ and rNS1^RUN-18^ mutants are indicated in red. The green diamonds mention the N-linked glycans at positions N130 and N207. The four amino-acid substitutions that differentiate rNS1^RUN-18^ from rNS1^DES14^ are indicated. In (**B**), immunofluorescence assays were performed using anti-6x(His) antibody (green) as the primary antibody. Nuclei were stained with DAPI (blue). Immunostained cells were visualized by fluorescence microscopy. The same magnification was used throughout (x100). In (**C**), immunoblot assay was performed on non-heated samples using anti-NS1 mAb D/2/D6/B7 (NS1 Ab^D/2/D6/B7^). The open arrowhead indicates glycosylated rNS1 protein ((rNS1)_2_). The gray arrowhead indicates non-glycosylated rNS1 protein (ng-(rNS1)_2_). The β-actin protein served as a protein loading control.

**Figure 7 ijms-22-01951-f007:**
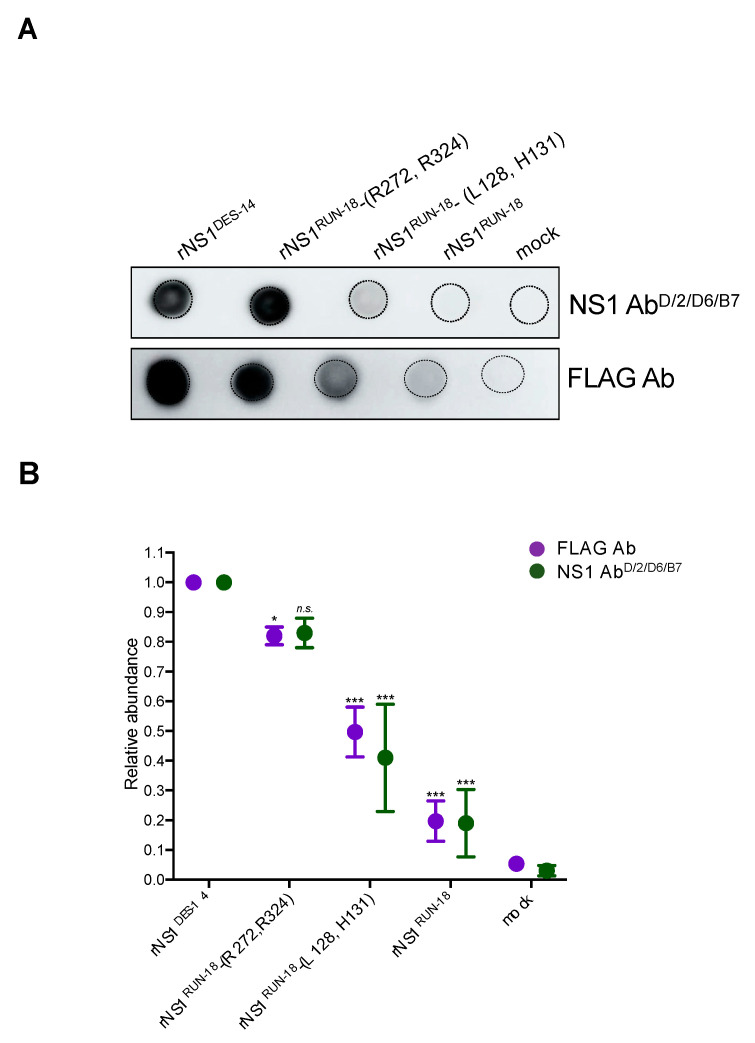
DENV-2 rNS1 secretion in Huh7 cells. Huh7 cells were transfected 48 h with plasmids expressing rNS1^DES-14^ or rNS1^RUN-18^ and its two mutants, rNS1^RUN-18^-(L128, H131) and rNS1^RUN-18^-(R272, R324), or mock-transfected (mock). In (**A**), samples of cell supernatants were analyzed by dot-blotting using anti-NS1 mAb D/2/D6/B7 (NS1 Ab^D/2/D6/B7^) or anti-FLAG antibody (FLAG Ab). In (**B**), signal intensity was quantified using the Image J software to evaluate the amount of secreted rNS1. The secretion of rNS1^RUN-18^ wild-type or its mutants is relative to rNS1^DES-14^. The results are the mean of three independent assays. *p*-values were determined on the comparison of rNS1^RUN-18^ and its mutants with rNS1^DES-14^ (*** *p* < 0.001, * *p* < 0.1; *n.s.*: not significant).

**Figure 8 ijms-22-01951-f008:**
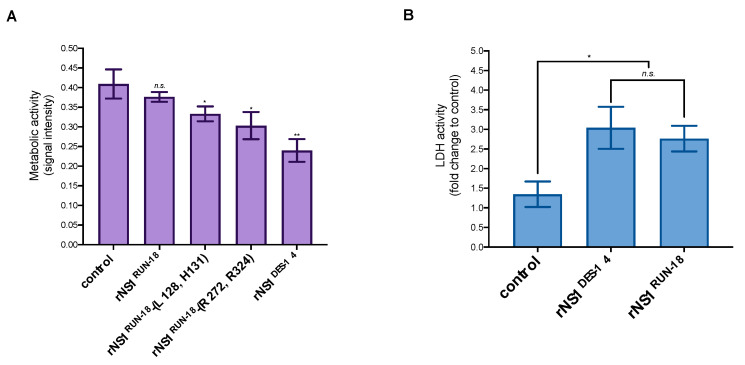
Effects of SWIO DENV-2 rNS1 expression on Huh7 cells. Huh7 cells were transfected 48 h with plasmids expressing rNS1^DES-14^ or rNS1^RUN-18^ and its two mutants, rNS1^RUN-18^-(L128, H131) and rNS1^RUN-18^-(R272, R324), or mock-transfected (control). A plasmid expressing reporter protein GFP was used as a control (control). In (**A**), cell metabolic activity was measured using an MTT assay. The results are the mean (±SEM) of eight replicates. Pairwise comparisons of rNS1 constructs with control were performed and noted (** *p* < 0.01; * *p* < 0.05; *n.s*. not significant). In (**B**), a measure of LDH activity in supernatants of Huh7 cells expressing rNS1. The LDH activity was expressed as a fold change relative to mock-transfected cells. *p*-values were determined for rNS1^DES-14^ or rNS1^RUN-18^ vs. control or rNS1^DES-14^ vs. rNS1^RUN-18^. The results are the mean (±SEM) of six replicates (* *p* < 0.05, *n.s.*: not significant).

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
