# Peer review of "Instability of the NS1 Glycoprotein from La Reunion 2018 Dengue 2 Virus (Cosmopolitan-1 Genotype) in Huh7 Cells Is Due to Lysine Residues on Positions 272 and 324"

_ijms, 2021, doi:10.3390/ijms22041951_

Round 1

Reviewer 1 Report

Line 34: important restrictions? Can you tell this specifically ? What is restricted and why so ?

Line 45-56: I don't get a point what authors try to explain to us ? Is Reunion/2018 DENV-2  a rare subtype ?

Line 73-86: This paragraph is the most important part in introduction. However, it is not clear why authors focus on this DENV-2 genotype. It is difficult to justify why variations of four amino acids in glycoproteins are expected to cause biological difference of NS1. The authors must explain what is a reason to focus on these four amino acid variation. 

Line 85: How do NS1-272 and-324 affect dimerization and secretion ?  As described in line 28-29, do mutations of these two residues cause the defect of NS1 expression and how ? 

Line:161-162: The amount of NS1 monomer in Run-18 is less than DES-14. Therefore, the effect of NS1 to cause cell death (LDH release) is not depending on the amount of NS1 (nor stability). The role of NS1 in inducing cell death is unclear from this result. 

Line 181: LDH is one of the marker to guess multiple organ failures, especially liver and muscle at the time of shock. The severe dengue symptoms are generally related to severe viremia or quite large amounts of virus titers that are reflected as a result of plentiful dead cells, by which liver failure and high LDH in the serum appear. Figure 4 shows NS1 induced LDH release, although if you conclude NS1 is the cause for cell death, you need to express other viral protein(s) and show that other viral proteins don't induce LDH release. This experiment is not promising the author's point.

Line 194: "a brighter immunostaining" cannot be accepted to explain the result scientifically. Can you do the quantitation in immunostaining ? 

Line 221- paragraph:

Figure 6 didn’t show the ratios of dimer vs monomer among DES-14, Run-18, and Run18 two mutants. What about that ratio among those ? It seems the total amount of dimer are improved in Run18 two mutants, although this data couldn’t couldn’t conclude any, since the important point must be the ratio according to your story.  

Line 222-223: quantitation and statistical analysis is required.

Line 227-: I don’t get a point why authors decide to make mutations Asn to Glu on 130 and 207.  

Line 270-271: Can you show the quantitation statistically from the dot blot data ? I think that the dot blot data is not reliable, compared with western blot. You need to explain why you chose dot blot method.

Line 315-316: How was this part analyzed ? "3D prediction" is not enough to explain the method.

Line 331: Does "slightly higher level" mean a significant difference statistically ?

Line 338: Does "low production" mean quantitated by comparison and analyzed statistically ?

Line 350: A reference is needed.

Line 355: How impacted ? increased or reduced the amounts ?

Line 376: Does "weak efficacy" mean significantly by statistic calculation ?

Author Response

January 25th 2021

Ref : ijms-1066725

Dear Editors,

Thank you for your high consideration of our submitted manuscript ijms-1066725. Please in the revised document our point-by-point responses to the reviewer’s comments. A brief summary of the changes made to the article can be found on this page.

The major changes made to the article are the following:

  • The title has been modified.
  • The section INTRODUCTION has been clarified.
  • The section DISCUSSION has been reshaped.
  • The data of Figure 4 have been fused to Figure 6A to generate a new Figure 4A in the revised version of manuscript.
  • Quantitation and statistical analysis have been performed in new Figures 4B and 7B.
  • We have generated a new Figure 8 showing data from MTT and LDH assays.
  • We provided a new supplementary Figure S2.

We greatly appreciate the efforts of the two reviewers and your editorial team of International Journal of Medical Sciences to help us make this the strongest work, and we look forward to your reply.

Sincerely yours,

Dr. Marjolaine ROCHE & Pr. Philippe DESPRES

corresponding authors

La Reunion island University and UM 134 PIMIT

Platform CYROI, 97491 Saint Clotilde France

[email protected]

[email protected]

Response to reviewers:

# Reviewer 1

Line 43: important restrictions? Can you tell this specifically? What is restricted and why so?

From the reviewer comment, we understood that the sentence might be unclear for a reader. Consequently, the text has been modified in lines 42-45 of the revised version of the manuscript.

Line 45-56: I don't get a point what authors try to explain to us? Is Reunion/2018 DENV-2 a rare subtype?

From the reviewer comment, we understand that the information on the genotype of DENV-2 strains used in the study might be too poor for a reader. Consequently, the text has been modified in lines 46-57 of the revised version of the manuscript.

Line 73-86: This paragraph is the most important part in introduction. However, it is not clear why authors focus on this DENV-2 genotype. It is difficult to justify why variations of four amino acids in glycoproteins are expected to cause biological difference of NS1. The authors must explain what is a reason to focus on these four amino acid variations. 

From the reviewer comment, we understand that rationale of our study might be unclear for a reader. As proposed by the reviewer, the text has been modified in lines 74-87 of the revised version of the manuscript.

Line 85: How do NS1-272 and-324 affect dimerization and secretion?  As described in line 28-29, do mutations of these two residues cause the defect of NS1 expression and how? 

From the reviewer comment, we understood that the sentences might be unclear for a reader. Consequently, the text has been clarified in the revised version of the manuscript in lines 87-89. The abstract has been modified in lines 26-29 of the revised manuscript accordingly

Line:161-162: The amount of NS1 monomer in Run-18 is less than DES-14. Therefore, the effect of NS1 to cause cell death (LDH release) is not depending on the amount of NS1 (nor stability). The role of NS1 in inducing cell death is unclear from this result. 

Line 181: LDH is one of the markers to guess multiple organ failures, especially liver and muscle at the time of shock. The severe dengue symptoms are generally related to severe viremia or quite large amounts of virus titers that are reflected as a result of plentiful dead cells, by which liver failure and high LDH in the serum appear. Figure 4 shows NS1 induced LDH release, although if you conclude NS1 is the cause for cell death, you need to express other viral protein(s) and show that other viral proteins don't induce LDH release. This experiment is not promising the author's point.

We appreciate these interesting points raised by the reviewer #1. We have generated a new Figure 8 showing the consequences of SWIO DENV-2 rNS1 expression on MTT and LDH activity in Huh7 cells. We showed that cellular metabolism (mitochondrial activity) is affected by expression of DES-14 rNS1 but not RUN 18 rNS1. However, a comparable loss of cell viability (LDH release) was observed in response of DES-14 rNS1 or RUN-18 rNS1 expression indicating that the greater amount of soluble DES-14 rNS1 in Huh7 cell supernatant relates to a higher protein expression level but not to an increased loss of cell integrity. In conclusion, rNS1DES-14 and rNS1RUN-18 strongly differ in both intra and extracellular expression level, with correlated consequences on cellular metabolism but without significant difference in cell viability (at least in the 48 h time window). These points have been clarified in lines 288-304 of the revised version of the manuscript.

Line 194: "a brighter immunostaining" cannot be accepted to explain the result scientifically. Can you do the quantitation in immunostaining ? 

We agree with the reviewer #1. So far no quantitation in immunostaining has been performed and we apologize for not being able to provide this information in the current manuscript. Consequently, this assumption has been deleted accordingly in the revised manuscript. The text has been modified in 197-199 of the revised manuscript accordingly.

Figure 6 didn’t show the ratios of dimer vs monomer among DES-14, Run-18, and Run18 two mutants. What about that ratio among those? It seems the total amount of dimer are improved in Run18 two mutants, although this data couldn’t conclude any, since the important point must be the ratio according to your story.  

We appreciate this important point raised by the reviewer #1. The suggested experiments would provide valuable information on the expression level of DENV-2 rNS1 in Huh7 cells. Consequently, the immunoblotted NS1 monomeric forms were added in the new Figure 4A in the revised version of the manuscript The Figures 4 and 6 have been substituted by a new Figure 4B showing the protein expression levels of rNS1 monomeric and dimeric forms from DES 14 as well as RUN-18 and its mutants. The relative abundances of rNS1 monomeric and dimeric forms have been determined for each SWIO DENV-2 rNS1 constructs (Figure 4B). The text has been modified in lines 171-175 and 209-222 of the revised manuscript accordingly.

Line 221- paragraph:

Line 222-223: quantitation and statistical analysis is required.

We agree with the reviewer #1. A new Figure 4B showing the protein ratio between the dimer NS1 and house-keeping protein has been introduced in the revised version of the manuscript. The text has been modified in lines 171-175 and 210-223 of the revised manuscript accordingly.

Line 227: I don’t get a point why authors decide to make mutations Asn to Glu on 130 and 207.  

We appreciate this point raised by the reviewer #1. A rationale explaining the interest of non-glycosylated DENV-2 rNS1 mutants bearing the amino-acid changes N13Q and N207Q has been introduced in lines 225-230 of the revised version of manuscript.

Line 270-271: Can you show the quantitation statistically from the dot blot data? I think that the dot blot data is not reliable, compared with western blot. You need to explain why you chose dot blot method.

We appreciate this point raised by the reviewer #1. The new Figure 7 shows in (A) the accumulation of soluble rNS1 in Huh7 cell supernatant by dot-blot assays and in (B), quantitation of signal intensity from dotted samples with statistical analysis. The text has been modified in lines 263-277 of the revised manuscript accordingly.

Line 315-316: How was this part analyzed? "3D prediction" is not enough to explain the method.

We agree with the reviewer #1. A such information has been removed in the revised version of the manuscript.

Line 331: Does "slightly higher level" mean a significant difference statistically?

We appreciate this point raised by the reviewer #1. The new Figure 4B provides information on the protein level expression of RUN-18 NS1 mutant bearing the mutations P128L and Q131H. The text has been modified in lines 339-342 and 352-355 of the revised manuscript accordingly.

Line 338: Does "low production" mean quantitated by comparison and analyzed statistically?

We appreciate this point raised by the reviewer #1. The new Figure 7B provides information on the relative abundance of soluble RUN-18 NS1 in Huh7 culture cell supernatant. The text has been modified in lines 352-355 and 387-391 of the revised manuscript accordingly.

Line 350: A reference is needed.

The complete references have added in the section “References” and mentioned in supplemental Table 1 in the revised version of the manuscript. The text in lines 359 to 368 has been reshaped in the revised version of the manuscript make the message clearer.

Line 355: How impacted? increased or reduced the amounts?

From the reviewer comment, we understood that the sentences might be unclear for a reader. The text has been reshaped in lines 368-370 and 372-374 of the revised manuscript accordingly.

Line 376: Does "weak efficacy" mean significantly by statistic calculation?

We appreciate this point raised by the reviewer #1. The new Figure 7B provides information on the secretion rate of soluble RUN-18 NS1 in Huh7 cells.

Reviewer 2 Report

Title: Natural mutations affect secretion of NS1 glycoprotein from dengue 2 virus Cosmopolitan genotype in South West Indian ocean

Summary:

The Authors investigated whether four amino acid changes in NS1RUN-18, compared to NS1DES-14, elicited a difference in protein dimerization and secretion. They engineered the 4 changes into an NS1DES-14 open reading frame and expressed proteins in different cultured cell lines. They examined expression levels and antibody reactivity via western blot of lysates from cells transfected to express the proteins. They also examined dimer formation, cell secretion and cell viability. They conclude from their results that two lysine residues play an important role in expression from a human liver cell line.

The manuscript is generally well-written and the study is interesting. NS1 is clearly an important protein in DENV virus infection and pathogenesis. The experiments were generally well designed and described. Enthusiasm was dampened somewhat by the lack of explanation for quantitation of transfection efficiency and protein expression in western blot and immunofluorescence.

Comments:

  1. The title uses the phrase “Natural mutations”. I’m not sure that is the best way to phrase these changes, or polymorphisms, in the amino acid sequence. Mutations compared to which sequence? This is a tricky issue with RNA viruses. The Authors can consider changing the title but it is not critical since the use of the word “mutation” is not strictly governed in most published work.

  1. It is generally unclear how the protein bands in the western blots are quantified. There is a beta-actin control but is some method used to quantitate dimer or monomer formation using the controls? There is also no description of how many times the western blot experiments were independently repeated.

In Figure 4A, little dimer is detected for rNS1RUN-18 and it is concluded that there is a defect in protein stability or dimerization in Huh7 cells. However, what if the transfection was less efficient for the rNS1RUN-18 plasmid and there is simply less monomer and therefore dimer? Is there a method to evaluate transfection efficiency? If not, was quantitation of monomer versus dimer bands performed?

  1. In Figure 5, it is indicated that the rNS1RUN-18 (R272 R234)-expressing cells presented a brighter immunostaining as compared to cells expressing other versions. How was this quantitated? A difference in brightness of immunostaining was not readily apparent from the images.

Minor issues:

  1. Line 183. “not impact” should be “no impact”
  2. Line 273. Is “rNS1RUN-18 (R272, R234)” correct? Should it be rNS1DES-14 (R272 R324)?
  3. Lines 315-316. The sentence here is awkward and a little unclear and should be rewritten.
  4. Line 328. Should “rise the hypothesis” be “raises the hypothesis”?

Author Response

January 25th 2021

Ref : ijms-1066725

Dear Editors,

Thank you for your high consideration of our submitted manuscript ijms-1066725. Please in the revised document our point-by-point responses to the reviewer’s comments. A brief summary of the changes made to the article can be found on this page.

The major changes made to the article are the following:

  • The title has been modified.
  • The section INTRODUCTION has been clarified.
  • The section DISCUSSION has been reshaped.
  • The data of Figure 4 have been fused to Figure 6A to generate a new Figure 4A in the revised version of manuscript.
  • Quantitation and statistical analysis have been performed in new Figures 4B and 7B.
  • We have generated a new Figure 8 showing data from MTT and LDH assays.
  • We provided a new supplementary Figure S2.

We greatly appreciate the efforts of the two reviewers and your editorial team of International Journal of Medical Sciences to help us make this the strongest work, and we look forward to your reply.

Sincerely yours,

Dr. Marjolaine ROCHE & Pr. Philippe DESPRES

corresponding authors

La Reunion island University and UM 134 PIMIT

Platform CYROI, 97491 Saint Clotilde France

[email protected]

[email protected]

Reviewer 2

The manuscript is generally well-written and the study is interesting. NS1 is clearly an important protein in DENV virus infection and pathogenesis. The experiments were generally well designed and described. Enthusiasm was dampened somewhat by the lack of explanation for quantitation of transfection efficiency and protein expression in western blot and immunofluorescence.

Comments:

  1. The title uses the phrase “Natural mutations”. I’m not sure that is the best way to phrase these changes, or polymorphisms, in the amino acid sequence. Mutations compared to which sequence? This is a tricky issue with RNA viruses. The Authors can consider changing the title but it is not critical since the use of the word “mutation” is not strictly governed in most published work.

We appreciate this point raised by the reviewer #2. The title has been changed in the revised version of the manuscript accordingly.

  1. It is generally unclear how the protein bands in the western blots are quantified. There is a beta-actin control but is some method used to quantitate dimer or monomer formation using the controls? There is also no description of how many times the western blot experiments were independently repeated.

We agree with the reviewer #2. This have been clarified in the legend of new Figure 4 which includes information on the protein expression levels of rNS1 monomeric and dimeric forms in Huh7 cells. The sentences have been modified in lines 171-175 and 209-222 of the revised version of the manuscript accordingly.                             

In Figure 4A, little dimer is detected for rNS1RUN-18 and it is concluded that there is a defect in protein stability or dimerization in Huh7 cells. However, what if the transfection was less efficient for the rNS1RUN-18 plasmid and there is simply less monomer and therefore dimer? Is there a method to evaluate transfection efficiency? If not, was quantitation of monomer versus dimer bands performed?

We appreciate this point raised by the reviewer #2. The new Figure S2 demonstrates that efficiency of transfection is similar for the two plasmids expressing either RUN-18 rNS1 or DES-14 rNS1. A sentence has been added in lines 163-165 of revised version of the manuscript. As shown in new Figure 4B, quantitation of rNS1 monomeric and dimeric forms has been performed. The text has been modified in the revised version of the manuscript accordingly.

  1. In Figure 5, it is indicated that the rNS1RUN-18(R272 R234)-expressing cells presented a brighter immunostaining as compared to cells expressing other versions. How was this quantitated? A difference in brightness of immunostaining was not readily apparent from the images.

We agree with the reviewer #2. So far, no quantitation in immunostaining has been performed and we apologize for not being able to provide this information in the current manuscript. Consequently, this assumption has been deleted accordingly in the revised manuscript. The text has been modified in lines 197-199 of the revised version of the manuscript accordingly.                      

Minor issues:

  1. Line 183. “not impact” should be “no impact”

Please accept our apologize for this typographical error. We address a remark to the reviewer #2 that the lines 288-304 have been rewritten in the revised version of the manuscript according to the comments of reviewer #1.

  1. Line 273. Is “rNS1RUN-18 (R272, R234)” correct? Should it be rNS1DES-14 (R272 R324)?

Please accept our apologize for this mistake. This error has been corrected in the new Figure 7A.

  1. Lines 315-316. The sentence here is awkward and a little unclear and should be rewritten.

We agree with the reviewer #2. The information has been deleted in the revised version of the manuscript accordingly

  1. Line 328. Should “rise the hypothesis” be “raises the hypothesis”?

Please accept our apologize for this grammatical error. The correction has been made in the revised version of the manuscript accordingly.

Round 2

Reviewer 1 Report

Line 147: Based on the result from Fig.3, it is apparent that rNS1-RU18 is expressed as similar amounts comparable to rNS1-DES-14 (Flag antibody showed), although  the thing that the serum antibody did not means that the antibody doesn't bind well to the RU18, due to the amino acid alteration and 3D structure changed.  That is why you just need to say the structure was impacted., not the stability at this point.

Line 160:  I don't think that this is the defect for the expression. How can you explain the coding sequence affect the expression ? I think the stability or half life of the protein can be changed.  

Line 198: I don't think you can compare slight difference of the protein stability in IFA. What is your point in this sentence and how can you explain RUN-18 from IFA results? What do you mean properly expressed ? 

The usage of "expression" is confusing, since you are just looking at the protein amounts after protein expression processes including posttranslational modification at ER membrane etc. and there must be cellular regulation between protein expression and degradation. Amino acid change can cause the change of degradation degree affecting half life.  

As all results, it seems stability seems affected by the 4 mutations and therefore secreted amounts are reduced. I still don't get a clear explanation from your sentences. If you insist "expression is different", you need to show the mRNA amounts between two NS1s.

Author Response

Ref : ijms-1066725R1

February 9th 2021

Dear Editors,

Thank you for your high consideration of our submitted manuscript ijms-1066725R1. Please in the revised document our point-by-point responses to the last comments by the reviewer #1.

We greatly appreciate the efforts of the reviewer #1 and your editorial team of International Journal of Medical Sciences to help us make this the strongest work, and we look forward to your reply.

Sincerely yours,

Dr. Marjolaine ROCHE & Pr. Philippe DESPRES

corresponding authors

La Reunion island University and UM 134 PIMIT

Platform CYROI, 97491 Saint Clotilde France

[email protected]

[email protected]

Responses to reviewer #1:

We greatly thank Reviewer #1 for his/her deep reviewing that helped us to improve our manuscript quality.

Line 147: Based on the result from Fig.3, it is apparent that rNS1-RU18 is expressed as similar amounts comparable to rNS1-DES-14 (Flag antibody showed), although the thing that the serum antibody did not means that the antibody doesn't bind well to the RU18, due to the amino acid alteration and 3D structure changed.  That is why you just need to say the structure was impacted., not the stability at this point.

We agree with the reviewer’s comment. In accordance, the sentence in lines 147-148 has been modified in the revised version of the manuscript accordingly.

Line 160:  I don't think that this is the defect for the expression. How can you explain the coding sequence affect the expression ? I think the stability or half life of the protein can be changed.  

We agree with the reviewer’s comment. In accordance, the sentence in lines 160-162 has been modified in the revised version of the manuscript accordingly.

Line 198: I don't think you can compare slight difference of the protein stability in IFA. What is your point in this sentence and how can you explain RUN-18 from IFA results? What do you mean properly expressed ? 

We agree with the reviewer’s comment. In accordance, the sentence in lines 197-198 has been modified in the revised version of the manuscript accordingly.

The usage of "expression" is confusing, since you are just looking at the protein amounts after protein expression processes including posttranslational modification at ER membrane etc. and there must be cellular regulation between protein expression and degradation. Amino acid change can cause the change of degradation degree affecting half life.  

As all results, it seems stability seems affected by the 4 mutations and therefore secreted amounts are reduced. I still don't get a clear explanation from your sentences. If you insist "expression is different", you need to show the mRNA amounts between two NS1s.

We appreciate the reviewer's careful examination of our data and we agree that the usage of “expression” could lead to some ambiguity for the reader. In accordance with the reviewer’s comment, we modified the main text to make it clearer. Consequently, the manuscript was tune out at several places to better acknowledge the fact that RUN-18 NS1 expressed in Huh7 cells was poorly stable compared to DES-14 NS1. Changes have been introduced (underlined in blue) throughout the main text: lines 26-30 (Abstract), lines 87-89 (Introduction), lines 170-172 (Results), lines 175-76 (Results), 197-198 (Results), lines 211-214 (Results), lines 220-221 (Results), lines 223-224 (Results), lines 232-234 (Results), lines 260-271 (Results), lines 333-335 (Discussion), lines 345-351 (Discussion), 362-366 (Discussion), and 381-387 (Discussion). In accordance with the general comments by the reviewer#1, the title of the manuscript has been modified.

Reviewer 2 Report

Comments:

In general, the Authors have done a satisfactory job addressing most concerns raised in the previous version of the manuscript.

  1. The Authors added an immunofluorescence experiment (Figure S2) to indicate transfection efficiency of the plasmids was similar. For transfection efficiency, they should calculate the percentage of cells expressing the versions of NS1. They can count total cells via the DAPI stain. Otherwise, % expressing cells can be quantitated using flow cytometry. There should be some quantitation or estimation of transfected cell numbers otherwise the experiment added is not adequate.

Minor Comments:

The text added to the revision should be examined for clarity, a few examples below.

  1. Line 264. “We prompted us to determine if…” is awkward and should be rewritten.

  1. Line 341. “A such result prompted us to understand…” is awkward and should be rewritten.

Author Response

February 9th 2021

Ref : ijms-1066725

Dear Editors,

Thank you for your high consideration of our submitted manuscript ijms-1066725. Please in the revised document our point-by-point responses to the new comments by the reviewer #2. A brief summary of the changes made to the article can be found on this page.

A major change made to the article is the following:

  • We have modified the Figure S2 with S2A and new S2B.

We greatly appreciate the efforts of the reviewer and your editorial team of International Journal of Medical Sciences to help us make this the strongest work, and we look forward to your reply.

Sincerely yours,

Dr. Marjolaine ROCHE & Pr. Philippe DESPRES

corresponding authors

La Reunion island University and UM 134 PIMIT

Platform CYROI, 97491 Saint Clotilde France

[email protected]

[email protected]

Responses to reviewer #2:

QUERY #1.

We agree with the reviewer. The Figure S2 has been modified with the counting of transfected Huh7cells expressing RUN-18 rNS1 or DES-14 rNS1 by IF analysis. The new Figure S2B shows the percentages of NS1-positive cells at 24 h post-infection. No significant difference has been observed between RUN-18 rNS1 and DES-14 rNS1. The text has been modified accordingly at lines 164-166 and 473-476.

 QUERY #2.

 Please accept our apologize for the grammatical errors. The changes have been made in the revised version of the manuscript accordingly.
